

# Tipping dynamics in packaging systems: How a bottle reuse system was established and then undone

Mila K. F. Ong[1], Fenna Blomsma[1], Timothy M. Lenton[2]

[1] Faculty of Business, Universität Hamburg, Economics and Social Sciences, Hamburg, Germany
[2] Global Systems Institute, University of Exeter, Exeter EX4 4QE, UK

*Correspondence to*: Mila Ong (kimchau.ong@studium.uni-hamburg.de)

**Abstract.** In this paper we investigate the initially successful transition from regional bottle reuse for mineral water to a widespread bottle reuse system in Germany, its subsequent destabilisation, and what this teaches us about tipping dynamics in packaging systems. Our aim is to understand how the speed of sustainable change is influenced, focusing on key actors from

business and policymaking. Building on current research on positive tipping points, our case study demonstrates opportunities to create an enabling environment for change, the role of specific reinforcing feedback loops in accelerating sustainable transitions, and a successful business innovation and technology intervention. However, it also demonstrates the threat of destabilisation from the emergence of competing technologies, in this case single-use plastic bottles, and what we can learn from the unsuccessful business and policy efforts to stop the decreasing market share of reusable bottles. A failed policy

intervention illustrates the consequences of rushing into a flawed solution. We reflect on our findings considering current efforts to (re-)establish reuse systems as part of a transition towards a sustainable circular economy.

## 1 Introduction

To affect changes that involve socio-technical paradigm shifts, scholars traditionally reserve long time frames: with estimates ranging from 40–60 years for technological revolutions (Perez, 2011) up to 70 years for transitions to sustainable development

and innovation (Gross et al., 2018; Grin et al., 2010). This stands in stark contrast with societies needing urgent action on multiple pressing sustainability challenges, such as equality, education, resource scarcity, environmental degradation, pollution, and global warming.

This apparent contradiction has sparked interest in how positive change can be brought about faster. Socio-technical transition

research (Geels et al., 2017; Turnheim and Geels, 2013; Rosenbloom et al., 2020; Meckling et al., 2015) has already emphasised the potential for rapid and non-linear system change. Other work in this area ranges from research into fast product innovation cycles guided by purposeful learning (Weissbrod and Bocken, 2017; Antikainen et al., 2017), to reconceptualising innovation systems for deliberately accelerating the pace of change (Blomsma et al., 2022), to attempts to understand how relatively small interventions can lead to big changes through self-propelling feedback (Lenton et al., 2022).



However, the dynamics of rapid socio-technical change remains poorly understood. Comprehensive frameworks for empirically evaluating respective enabling conditions and triggers have only recently become a focus (Stadelmann-Steffen et al., 2021; Lenton et al., 2022; Fesenfeld et al., 2022; Winkelmann et al., 2022). Unanswered questions remain around the interaction between systemic conditions, actor agency and learning that causes change to accelerate or *tip* to become self-

sustaining. How to set direction towards sustainable outcomes needs addressing as current work has not advanced beyond recognising directionality challenges (Haddad et al., 2022; Bergek et al., 2023; Kemp et al., 2022). Equally, more research is required to determine how change can be made to endure (Sharpe and Lenton, 2021).

It's important to mindfully approach speed, as it may not always be beneficial. A speedy solution can be implemented, but still

not address a problem due to it not being fit-for-purpose (Sterner et al., 2010), or inadequately implemented (Howes et al., 2017). A fast solution can fail to endure or it can cause larger problems due to not engaging with other important aspects of the initial problem (Braun, 2002). Thus, mindlessly pursuing speed could ironically result in losing momentum: when the proposed change does not materialise as desired, resources may be unavailable for another attempt. Disillusionment may grow with calls for discarding the solution altogether, even when it was not the solution itself that was flawed, but rather the manner

of its implementation (Howes et al., 2017). For example, the revision of eco-energy-labelling in Germany to accommodate industry demands unintentionally diminished the effectiveness of a well-established labelling scheme by introducing new rating categories. The flaw was not the idea of modifying the labelling scheme, but the failure to simplify the implementation to successfully influence consumer choices (Heinzle and Wüstenhagen, 2012).

Hence, in this research we ask: "*What are the tipping dynamics that can bring about fast and sustainable change – and how can it be made to last*?". Our focus lies in examining the interplay between systemic conditions, business actors and policymaking in driving change. We investigate this through a historical case study of the tipping into a nationwide bottle reuse system in Germany, and the later tipping away from it into a less sustainable single-use bottle regime. This case was chosen because of the abrupt tipping towards a sustainable state[1], but also the subsequent failure to stabilise that state. The aim

is to improve the understanding of tipping dynamics, in particular those aimed at a rapid transition from a linear to circular economy.

---

[1] In this study we do not include impact as measured through LCA or Carbon-equivalent measures. There is little historical data to draw from and current metrics are not well equipped to provide a nuanced insight into this. Instead, we assume that when designed well, sufficiently mature and operating at scale, reuse systems have the potential to have less impact than single-use, disposable packaging due to the resource and energy savings they represent. We invite further work on the impact assessment of both systems.





## 1.1 The case study

The reuse pool system for mineral water bottles in Germany is well-established as the biggest reuse system in Europe and unique in its effectiveness and comprehensive scope. It is organised and managed by a business cooperative called *GDB* (*Genossenschaft Deutscher Brunnen*) and became successful with the introduction of a standardised pool bottle in 1969, that has remained unchanged since (Fig. 1). Reusable bottles held a relatively stable market share of over 80 % for three decades (1970–2000). Their environmental effectiveness is showcased by the high circulation rate of the bottles[2]: They can be reused

up to 50 times with average transportation distances of 260 km (DUH, 2014; UBA, 2016). The pool system ensures that the bottles are usually transported to the closest participating mineral water company and do not need to be returned to the original bottling company. The bottles are owned by the cooperative and lent to their business customers, noting that the vast majority of those hold an ownership stake as members of the cooperative (GDB, 2023d).

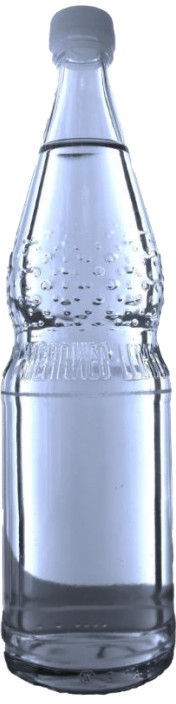

**Figure 1: The reusable glass pool bottle. Two rings (in the middle and at the bottom) function as "shock absorbers" to prevent breakage in the filling and cleaning process. The pearl-like patterns at the bottle neck expresses freshness while enabling a good grip (Bielenstein, 2019).**

---

[2] While environmental evaluations cannot deliver absolute certainty and are contingent upon various factors, a broad consensus exists on the sustainable breakeven point of reusable bottles to achieve less environmental impact then the single-use alternative, which is estimated at typically 10-15 circulations (DUH, 2014).





While historically reuse was a necessity driven by scarcity and resource constraints, with the advent of mass production and consumption, the focus shifted from reuse to disposability (König, 2019). This also affected the beverage industry with the introduction of single-use plastic bottles, leading to a destabilisation of the dominant reuse system with a rapidly decreasing market share of reusable bottles from over 80 % to around 40 % in the decade from 2000 to 2010 (Fig. 2).

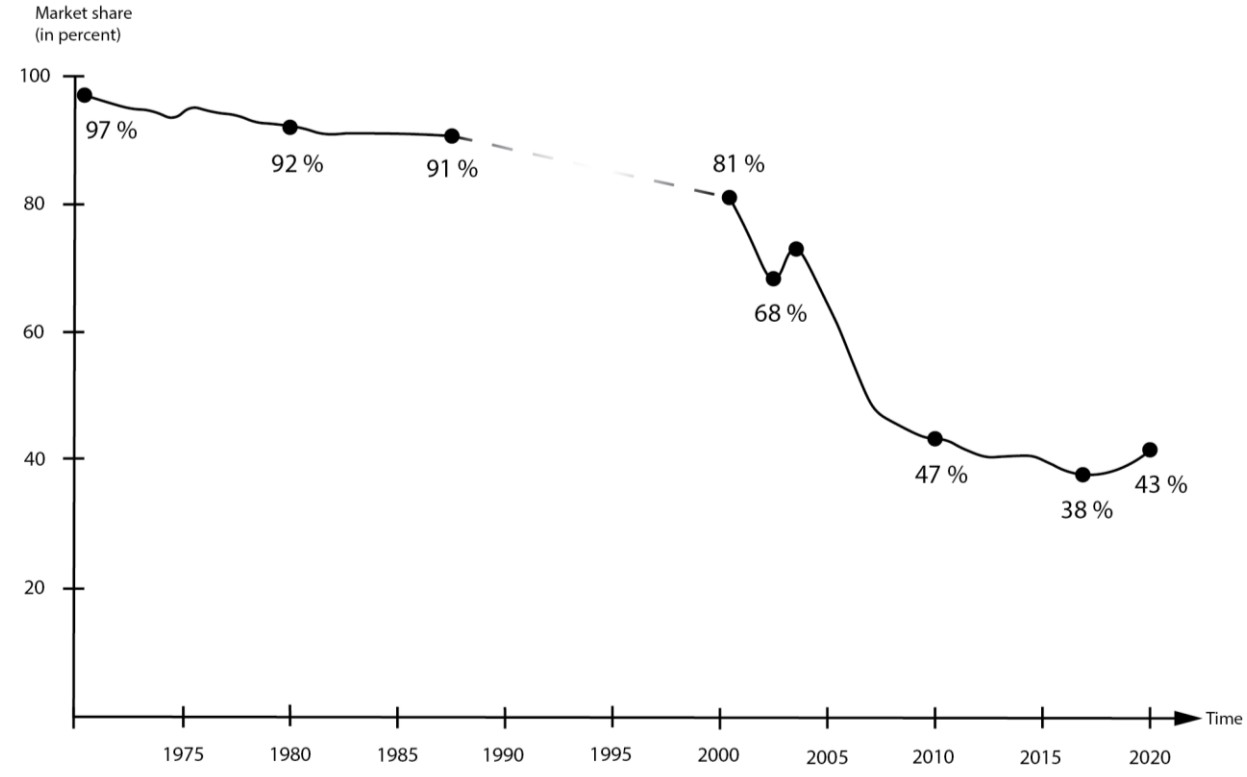

**Figure 2: Market share of reusable mineral water bottles (UBA, 1983, 2022)**

Today, the bottle reuse system coexists alongside a dominating single-use plastic bottle recycling system. While a deposit for single-use bottles (for beer, water, and soft drinks) was made mandatory in 2003 as a political intervention to stabilise the reuse market share, this only produced a temporary improvement of the reuse market share before a precipitous fall resumed.

We draw insights from these developments by investigating enabling and destabilising dynamics, identifying the feedback loops and interventions that propelled change and those that (temporarily) maintained stability to learn about business and policymaking opportunities for rapid change. We then reflect on what our findings mean for current efforts to (re-)establish and stabilise reuse systems for packaging, such as emerging industry initiatives and the EU Circular Economy agenda (European Commission, 2020). Our study also provides insight into the development of circular configurations – situations



where two or more circular economy strategies interact (Blomsma et al., 2023), in this case reuse and recycling strategies that
influence each other strongly over time with clear conflicts between them.

The paper is structured as follows. Section 2 introduces the underlying theoretical framework of temporal tipping dynamics in
socio-technical transitions. Section 3 outlines our research design. Section 4 presents our findings regarding two phases: first
the successful tipping to a widespread reuse system (1950–1985), and then the subsequent tipping away from the established
reuse system (1985–2010), followed by recent developments. Section 5 discusses the broader insights derived from this case
study, and Section 6 concludes.

## 2 Theoretical framework

### 2.1 Socio-technical transitions, systems, and pace of change

Sustainability transitions refer to the deliberate and systemic shifts in societies, economies, and industries towards more
sustainable and environmentally responsible practices, technologies, and systems (Geels, 2011; Smith et al., 2005; Stirling,
2009). Their systemic nature underscores the complexity and interconnectedness of the changes required, necessitating a
holistic approach that considers the interplay of various elements within societies, economies, and industries (Geels, 2011).

Transitions typically consist of many small, cumulative developments that culminate – over time – in the emergence of a new
regime. Although this may be accompanied by phases of acceleration, the accepted overall timeline is too long to achieve
targets like the SDGs and Paris Agreement (Gross et al., 2018; Grin et al., 2010; Kondratieff and Stolper, 1935).

However, there is also evidence that various actors can take actions to accelerate change (Sovacool, 2016; Victor et al., 2019)
and these can sometimes trigger the tipping of systems where change accelerates and becomes self-sustaining (Lenton, 2020).
Turnheim & Geels (2013) for example, argue that, within the context of sustainable transition policy, accelerating low-carbon
innovation necessitates the implementation of diffusion and deployment policies aimed at spreading and expanding niche
experiments beyond their initial boundaries, focusing on initiatives such as establishing new networks, promoting
entrepreneurial activities, setting standards, and enhancing education and training. Similarly, strategic niche management
(Schot and Geels, 2008) has been argued as pivotal for catalysing regime shifts, while acknowledging the importance of
linkages with ongoing external processes. Recently understanding of tipping dynamics has begun to be merged into the
transitions framework (Geels and Ayoub, 2023).

### 2.2 Understanding tipping dynamics – synthesis as the key to acceleration?

There are many models of tipping dynamics where change becomes self-propelling. The Diffusion of Innovation theory by
Rogers (1962), for example, identifies critical mass thresholds at which the rate of adoption of different innovations becomes



self-sustaining. Likewise, the model of increasing returns by Arthur (1989) identifies positive feedback mechanisms that can get strong enough to support self-propelling change, like economies of scale or technological reinforcement. Additionally, the coordination game considered by Kandori and colleagues (1993) describes how network effects resulting from coordination on a new technology enable a payoff that is superior to an incumbent technology.

Whilst these models have each demonstrated explanatory capacity, few synthesis efforts have been undertaken. To better understand how to bring about tipping it is useful to understand the differing roles of these different dynamics and how they interact. The Positive Tipping Points (PTPs) framework (Lenton, 2020; Sharpe and Lenton, 2021; FOLU and GSI, 2021; Lenton et al., 2022) attempts such a synthesis. Building on systems thinking and Meadows' 'leverage points' framework (Meadows, 1999; Abson et al., 2017) it highlights the importance of creating enabling conditions (e.g. price reductions or

shifts in social norms) before a small perturbation can trigger a socio-technical tipping point (Fig. 3), and understanding how feedback loops that can either accelerate or impede change (Lenton et al., 2022).

Central to PTP is understanding how a system can be deliberately tipped in a more desirable direction (Lenton, 2020). Specific actions, behaviours or policies can reach a critical threshold (Dakos et al., 2015; Kopp et al., 2016) that triggers transformative

system-wide change (Otto et al., 2020). When a small perturbation within a socio-technical system triggers a fundamental shift towards a qualitatively different state, strong positive feedback mechanisms are acting to amplify the effects of the small change. Once initiated, these dynamics can be abrupt and difficult to reverse. In some cases they may trigger a further chain reaction of change across sectors and scales, in a *positive tipping cascade* (Sharpe and Lenton, 2021; Geels and Ayoub, 2023).

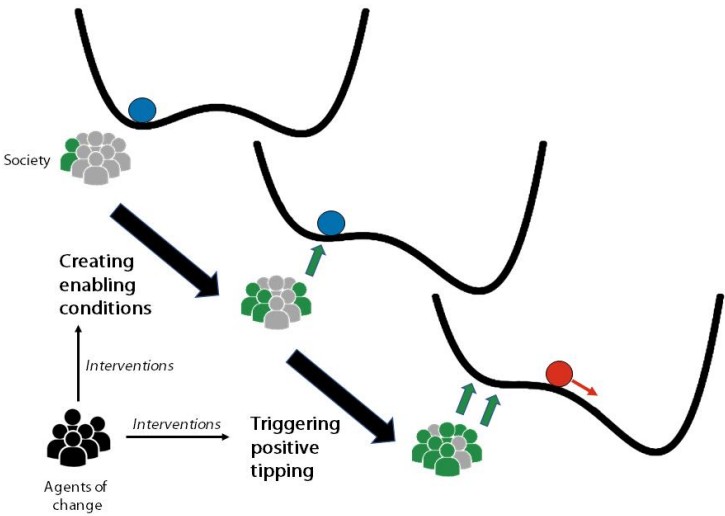

**Figure 3: A dynamical systems conceptualisation of positive tipping points (Lenton et al., 2022)**





In Lenton et al. (2022) enabling conditions are operationalised in a non-exhaustive list of variables involving population size, social network structure, information and capability, price, performance and quality, desirability and symbolism, accessibility and convenience, and complementarity (see also FOLU and GSI, 2021). Key reinforcing feedbacks that can support tipping dynamics are identified from synthesising different tipping point models, alongside interventions for different actors to trigger

tipping dynamics (Lenton et al., 2022; FOLU and GSI, 2021) (Fig. 4).

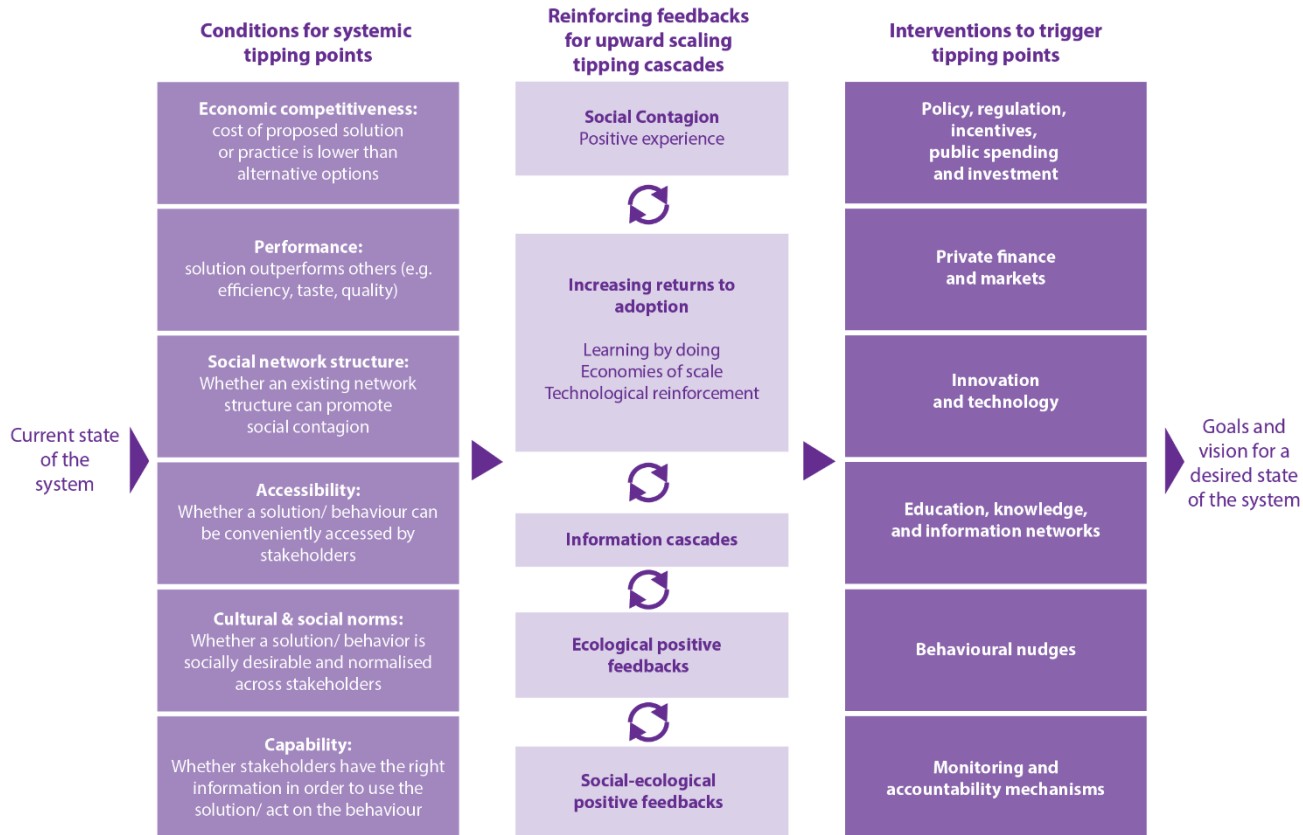

**Figure 4: Summary of framework for triggering positive tipping points, adapted from Lenton et al. (2022) and FOLU and GSI (2021).**

Thus far, the framework has been applied to the adoption of renewable energy and electric vehicles (Sharpe and Lenton, 2021), or policy changes that prioritise environmental protection (Fesenfeld et al., 2022; Tàbara et al., 2022). Previous work also identified potential social tipping interventions in subsystems like human settlements, financial markets, and education (Otto et al., 2020).



While existing PTP case studies tend to focus on significant changes in socio-technological systems from a macro-social perspective, there is potential to better understand agency by identifying precise actions for specific societal actors.

## 2.3 The role of business and policymakers as key actor groups

Sustainability-oriented transitions are aimed at addressing persistent environmental issues unlike past transitions that were primarily driven by emergent commercial opportunities (Geels, 2011). Hence, sustainability transitions can be more difficult

to make economically competitive compared to established technologies. Consequently, changes in economic governance, such as taxes, subsidies, and regulations are often necessary to facilitate the adoption of environmental innovations. Implementing such changes requires navigating politics and the involvement and reorientation of incumbent firms. These circumstances can cause (new) challenges due to varying and sometimes conflicting perspectives on the directionality of sustainability transitions (Stirling, 2009), the merits or drawbacks of specific solutions, and the most suitable policy

instruments or packages.

Previous research suggests the importance of considering synergistic effects and learning feedbacks among systems, which can lead to interconnected tipping dynamics across various domains (Stadelmann-Steffen et al., 2021). Both business actors and policymakers are pivotal agents of change, and their interactions can significantly impact the speed of transformative

change. Businesses have the entrepreneurial opportunities to engage in innovation and the necessary resources to drive change, while policymakers have the capacity to formulate regulatory frameworks and interventions that can trigger or impede social tipping dynamics (Otto et al., 2020; Smith et al., 2020; Sharpe and Lenton, 2021).

Yet, business actors and policymakers rarely agree over proposed political interventions. Businesses generally tend to oppose

or be sceptical towards policy interventions due to concerns about potential harm to their economic interests (Coen, 1997; Bernhagen, 2014). Criticism is also often directed towards existing policies for their perceived lack of comprehensiveness and strategic vision. Some argue that these policies primarily concentrate on exploiting marginal efficiency and innovation gains, rather than promoting a collaborative approach that encompasses a holistic transformation of socio-technical systems (Rosenbloom et al., 2020; Haddad et al., 2022). Conversely, policy interventions are necessary to address market failures, such

as externalities and imperfect information, to stimulate sustainable innovation and prevent possible competitive disadvantages, as well as to coordinate collective action between various stakeholders (Haddad et al., 2022; Bergek et al., 2023).

In our study, we therefore address a lack of research on how these business and policy interactions influence the speed of sustainable transitions by testing the applicability and interpretative power of the PTP framework (Fig. 4). We analyse specific

enabling and destabilising dynamics within our packaging system case to identify opportunities for action.



## 3 Research Design

Qualitative content analysis (Gioia, 2021) was used in our historical case study to address the following sub-questions: *Which tipping dynamics can be identified in this case that triggered or impeded the acceleration of change? (Where) were*

*opportunities taken or missed to enable and stabilise a widespread reuse system? What learnings does this case reveal for business actors and policymakers when it comes to the transition to a sustainable circular economy?*

To assess this case study, a comprehensive timeline of events was constructed based on secondary data, incorporating relevant information from the political and economic context, and developments within the mineral water industry. Industry information

was sourced from historical reports of a leading mineral water company *Gerolsteiner* (Lippert et al., 2012; Schuck, 2015) and the industry cooperative *GDB* (Bielenstein, 2019), supported by an expert interview. Additional insights were drawn from existing literature on the history of the mineral water industry (Eisenbach, 2004), as well as complementary literature on the German history of waste (König, 2019; Kleinschmidt and Logemann, 2021). To enrich the analysis, archival documents from the Federal Archive in Germany, specifically those relating to the beverage industry from 1980 to 1990, were consulted. Key

performance indicators like market shares and circulation rates were extracted from reports issued by the German Federal Environmental Agency (UBA, 1983, 2016, 2022). Some of these developments can only be presented at a surficial level, leaving more in-depth exploration for future research.

## 4. Results: The historical development of the German bottle reuse system

After placing the case in its historical context (up to the 1950s), this case study is divided into two phases, 1950–1985 and

1985–2010, followed by an elaboration of current developments since 2010 (see Fig. 5).

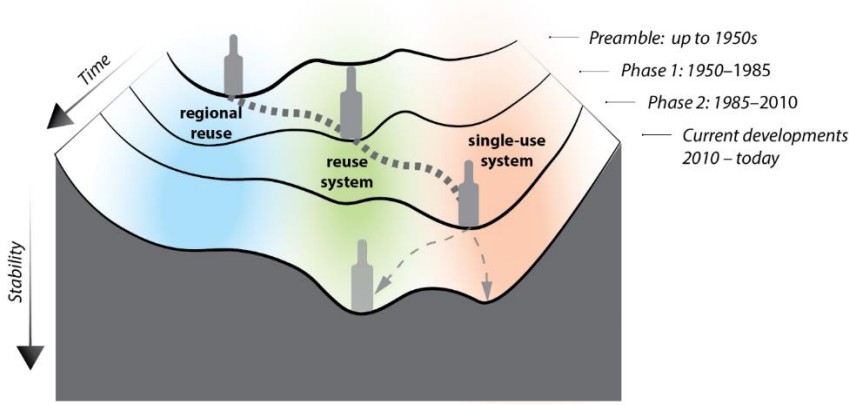

**Figure 5: Illustrative visualisation of the development of bottle waste management systems in Germany from regional reuse to a widespread reuse system to a single-use system, and potential future pathways. The valleys represent alternative stable states of the system, which are evolving over time. The bottle represents the actual state of the system at a particular time. The dashed line**
**shows the historical trajectory of the system and the dashed arrows the possible trajectories unfolding now and into the future.**



### 4.1. Preamble: Up to 1950s

Reuse behaviour had long been the standard before the "throwaway mentality" of modern society emerged. This was due to prevailing scarcity-driven economies, that made it necessary to maximise the exploitation of available resources and goods by
reusing, reutilising and repurposing them for as long as possible (Denton and Weber, 2022). Consequently, bottle reuse was a common procedure to save costs for mineral water companies, but there was no infrastructure for organised, large-scale reuse systems. Every mineral water company used its individually shaped bottles for reuse – leading to long, laborious, and expensive exchange and return processes – or directly discarded them through costly glass recycling (Eisenbach, 2004).

Already in 1875, a more efficient solution was suggested in the form of a uniform bottle design with removable labels. However, despite wide approval, implementing this proposal faced difficulties due to conflicting interests, and the ambitious project was eventually forgotten (Eisenbach, 2004). At the end of the nineteenth century, industry associations were established to face shared challenges and address common interests, contributing to a special social network structure within the industry. The first mineral water industry association was the *Verband Deutscher Mineralbrunnen* (*VDM*, founded in 1898) with the
aim of representing and protecting all German mineral water producers in the light of strong industry growth. (Lippert et al., 2012).

More importantly, in 1937 the *Genossenschaft Deutscher Brunnen* (*GDB*) was founded in close cooperation with the *VDM*. As a buying and selling cooperative they went beyond the tasks of conventional associations by also being responsible for the
procurement of everything a mineral water company needs for the production, bottling, transport, and marketing of its products (bottles, caps, machineries, etc.). To this day the *GDB* works with bottlers to draw up certain rules and ensure that they are adhered to. This helps to ensure that the quality of the bottles and transportation crates in circulation is always at a consistently high level, and that enough new bottles and crates are fed into the system (GDB, 2023e). The first product of the cooperative was a cola drink created in 1937 in reaction to the increasing popularity of *Coca-Cola* (GDB, 2023c).

The development of both associations was interrupted by the outbreak of World War II. After the war and the dissolution of National Socialist economic organisations, new approvals for associations and cooperatives were initially treated highly restrictively in the different occupied zones. After several efforts at reorganisation, in 1949 the *VDM* was re-founded as the overarching association for West Germany. In the Soviet occupation zone the establishment of business associations was
strongly restricted and most of the mineral water companies were nationalised or closed down (Eisenbach, 2004).

In parallel, the *GDB* cooperative reorganised itself in 1949 (GDB, 2023c) after the German economy, and subsequently the mineral water industry, gradually regained its strength as a consequence of a currency reform and financial assistance from the



Marshall plan. The economic upswing led to a rapid growth of the *GDB* cooperative with 133 members in the beginning of
the 1960s (Eisenbach, 2004), roughly three-quarters of all mineral water companies in West Germany.

## 4.2 Phase 1 (1950s – 1985): Tipping dynamics towards a widespread reuse system

Here, we describe the different dynamics (abbreviated as "R" for reinforcing feedbacks and "B" for balancing feedbacks) that
led to the destabilisation of the initial regional bottle reuse regime in West Germany and enabled tipping towards a widespread
reuse pool bottle system. In East Germany bottle reuse was already systematically organised in the 1950s by the existing waste
collection infrastructure known as *SERO* (*Sekundär-Rohstofferfassung*), which was part of the state-planned economy and
enabled citizens to sell their waste at a dense network of collection points for reuse or recycling (UBA, 1992). Legal standards
limited the variety of materials and shapes. However, as this system was detached from developments in West Germany and
ultimately could not endure after the German reunification in 1991, it falls outside the scope of our analysis.

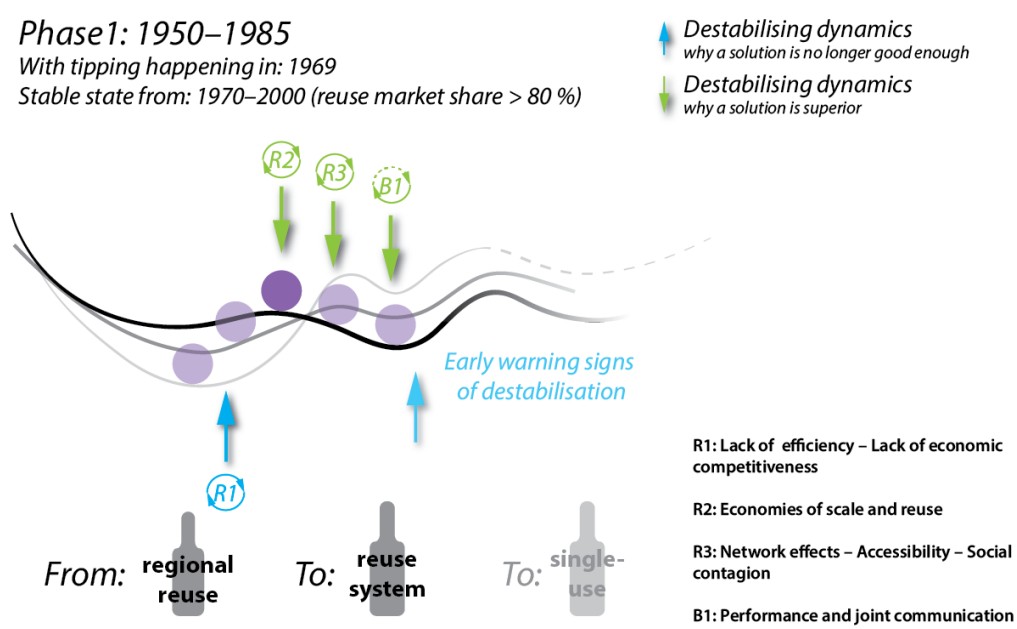

**Figure 6: Tipping dynamics from regional reuse to a widespread reuse system**

**R1: Lack of efficiency – Lack of economic competitiveness.** Approximately 210–150 million bottles and three million crates
were lost during World War II. Obtaining replacements was highly challenging due to material supply catastrophes and
frequent energy shortages in glass factories (Eisenbach, 2004). As bottles are an essential asset for beverage companies, this
led to the economic necessity to ensure the return and reusability of bottles. However, this was hindered by the inefficiency of





the regional reuse systems in West Germany. When *Coca-Cola* entered the market as a strong competitor, it became undeniable that the lack of efficiency in the prevailing regional reuse systems posed a competitive disadvantage (Eisenbach, 2004).

**R2: Economies of scale and reuse.** Given these shortcomings,
there was a need for an economically more reliable and sustainable solution to increase efficiency and to gain higher resilience. A standardisation of reusable bottles seemed plausible to save costs, since the production of standardised bottles in larger quantities promised cost reductions through
economies of scale. In addition, economies of reuse came into play, where the reuse of standardised bottles would help to spread the initial cost of production, resulting in a lower cost per use. Moreover, increasing reuse reduced the need for new bottle production, reducing overall production costs and material
needs. The resulting reduction of packaging waste and associated waste management costs contribute to a social-ecological positive feedback mechanism (reinforcing interaction between social intervention and ecological change). This later

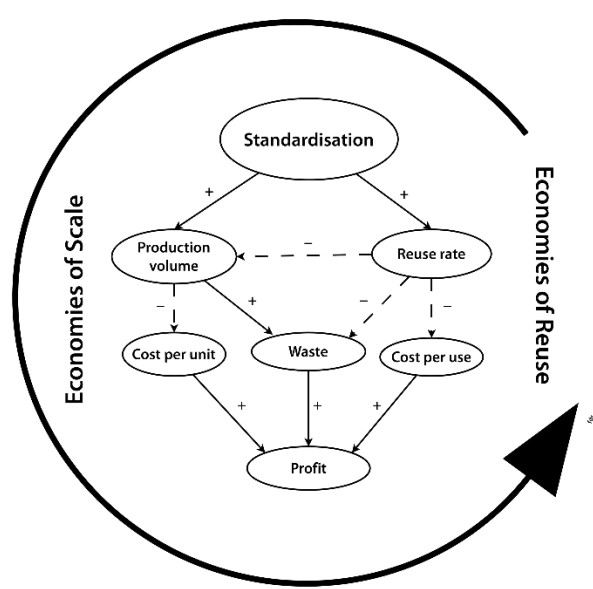

**Figure 7: Feedback loop of economies of scale and reuse**

became significant once it was recognised and acknowledged by environmental policymaking. However, first it became
evident that achieving a standardisation of bottles required the coordinated commitment of all competitors in the sector, which had not previously been feasible (Eisenbach, 2004).

**Business innovation and technology intervention: Introducing a bottle reuse pool system.** Finally in 1950, enabled by the existing social network structure of the industry associations, the *GDB* cooperative took the lead in developing a standardised
bottle and pushing towards a new regime. It commissioned the development of a uniform bottle shape, resulting in a standardised design guideline for a bottle with a lever cap already in 1950. However, this remained a niche experimentation and was not widely adopted due to technical impediments: The bottles still needed to be closed manually and were therefore unsuitable for machines. Additionally, the breakage rate of the caps and bottles was still high (Eisenbach, 2004).

With technology progressing, bottles with external screw caps were made possible approximately two decades later with significantly lower costs. This led to the investment of the *GDB* and *VDM* to develop a new standardised bottle in 1969. All those involved, including designers, market researchers, experts for glass works, and representatives of the mineral water companies and cooperatives, were aware that this bottle had the potential to become a relevant product in the long term and therefore the necessary steps needed to be carried out professionally and thoroughly to enable a sustainable design of the



product itself, and the system around it: The design of the bottle had to combine fulfilling technical needs with the requirements
of modern marketing, and a well-organised mechanism for the return and refill process through the *GDB* was essential
(Bielenstein, 2019).

After the revision of several designs and testing prototypes in market research, after five months the pool bottle design was
able to meet all requirements. The quick implementation of technical improvements, suggested from several market research
feedback cycles, was crucial to achieving a comprehensively well-performing and aesthetic design (see Fig. 1) (Bielenstein,
2019). The complementary technology of stackable and palletizable crates (which can be reused over 100 times) also played
a pivotal role in enabling smooth logistics and reinforcing the efficient performance of the system (Eisenbach, 2004).

**R3: Network effects – Accessibility – Social contagion.** By including all industry representatives and stakeholders in the
decision-making process and the final voting, a quick and almost industry-wide adoption of the pool bottle followed,
reinforcing the functioning of the reuse system as the system became more efficient the more companies participated
(Bielenstein, 2019). Additionally, the centralisation of the responsibility and management of the pool system by the *GDB*
cooperative enabled the streamlining of the procurement process for mineral water companies. This facilitated easier and more
reliable access to the necessary bottles as well as favourable pricing agreements, that were leveraged by the cooperatives
combined purchasing power (GDB, 2023a). This made participation in the *GDB* system even more attractive and beneficial
for the mineral water companies, leading to strong social contagion effects. To this day, around 95 % of all mineral water
companies are members of the GDB (GDB, 2023b). Through the established pool system, the companies were able to meet
their demands more promptly, presumably leading to increased productivity and customer satisfaction.


**Early warning signs of destabilisation.** Parallel to these developments in the mineral water industry, some major landscape
changes started to materialise after World War II. The stabilisation of the economy and political agenda resulted in an economic
boom (Fabian, 2021). This was further stimulated by liberal economic policies, such as reducing trade barriers, deregulating
industries, and promoting free-market principles. As prosperity increased and supply shortages were overcome, there was a
shift in spending and consumption habits: consumers became more selective and focused on their individual preferences
(Köhler, 2021; Fabian, 2021). While the pluralisation of product choices was a precondition for this shift (Fabian, 2021), it
was reinforced by the need of businesses to differentiate themselves in the market due to increasing competition. As the market
began to saturate in the 1960s, there was intense competition for customer loyalty and market share (Köhler, 2021). The
consumer goods industry responded by adapting tailored marketing strategies and expanding product ranges with personalised
products (Fabian, 2021; Beyering, 1987), which challenged the legitimacy of standardised packaging.

Simultaneously, the emergence of single-use packaging as a cost-effective alternative for preserving and transport of a variety
of goods marked a significant change in the way products were distributed and consumed. At first this did not affect the mineral



water industry due to taste impediments, other performance issues and prevailing reservations of consumers (Eisenbach, 2004).
Yet, there was growing awareness in the industry that the single-use bottle would eventually gain importance. Initial investments as early as 1967 have been recorded (Eisenbach, 2004) and the early permission to use plastic bottles for mineral water in France in 1968 caused a crucial impulse, prompting the forecast of the downfall of the reusable glass bottle in an industry magazine (Eisenbach, 2004).

A further significant development that increasingly posed new business challenges was the shift in the retail landscape. This was driven by significant changes and diversification in retail structures in the post-war decades, influenced by the prevailing self-service principle. During the 1970s, discounters, chain stores, and large specialist stores emerged, leading to a shift away from smaller independent retailers (Köster, 2021). Discounters focused on providing products at significantly lower prices compared to traditional retailers by limiting their store layout and product assortments and selling distributors' brands, offering
them at lower prices than established brands. This forced existing retailers to reevaluate their pricing strategies and become more competitive, putting pressure on their suppliers.

**B1: Performance and education.** From the end of the 1960s, single-use tin and aluminium cans were a greater threat to the beverage industry than single-use plastics, initiating campaigns that endorsed convenient use-and-dispose behaviour. A
counterplay of single-use proponents and opponents kept going, so that later in 1983 among other initiatives the association PRO MEHRWEG e.V. was founded. The initiative launched an informational campaign to raise public awareness around the environmental impact of single-use packaging (PRO MEHRWEG, 1984) and distributed information through radio and television features as well as numerous press publications. The joint communication of the mineral water industry was notably advanced, shaping the perception of the high quality of natural mineral water compared to table water (Eisenbach, 2004). This
and the, at that time, higher performance and desirability of reusable glass bottles presumably contributed to the stable market share of reusable bottles as single-use PET started to emerge.



**4.3. Phase 2: 1985 – 2010s: Tipping dynamics away from the established reuse system**

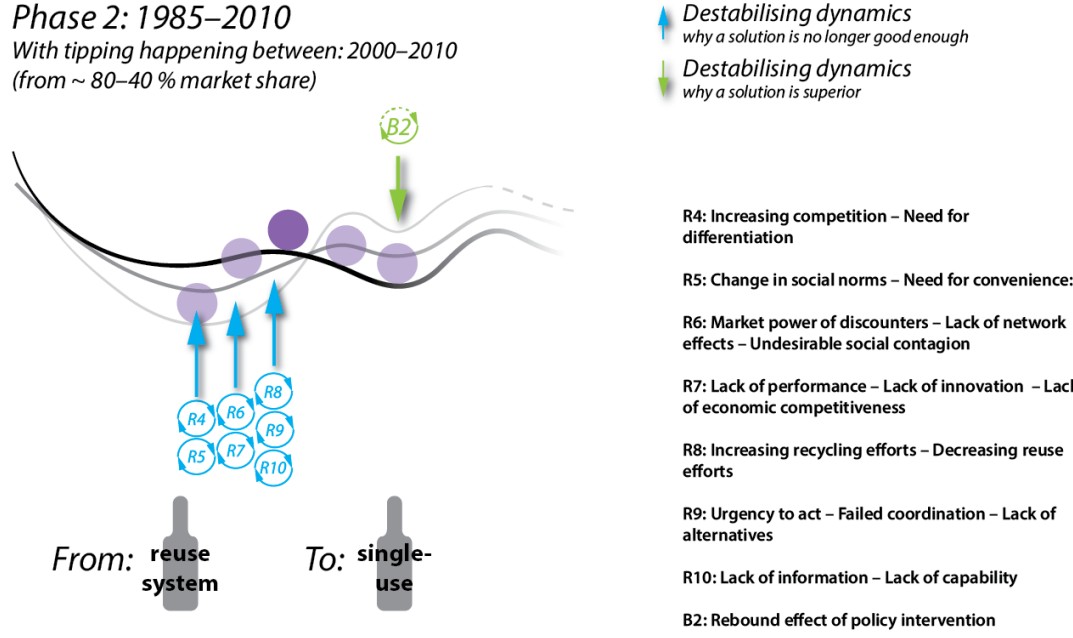


**Figure 8: Tipping dynamics from a widespread reuse system to single-use.**

**R4: Increasing competition – Need for differentiation.** Since the product "mineral water" itself does not leave much room for differentiation for marketing objectives, distinctive packaging design became more relevant to capture consumers'
attention. Further experimentations with new shapes aimed for a more luxurious and modern feel for use in settings like restaurants and venues (Lippert et al., 2012). The consequent shift towards using more individual reusable and single-use bottle shapes, instead of participating in the reuse pool system, made the latter less efficient (Lippert et al., 2012). Although individual bottle reuse solutions have contributed to the market share of reusable bottles, they dampened the network effects and economies of scale of the *GDB* bottle reuse pool system, since they can only be returned and used by one company.


**R5:  Change in social norms – Need for convenience.** The increasing need for convenience was determined by the rise of employment rates, small households, increasing average age of the population, increasing out-of-home consumption, decreasing time budget for shopping, food and beverage preparation, and decreasing willingness of households to reserve storage space for full and empty bottles (Fabian, 2021). Additionally, the promotion of single-use packaging as a luxury to



simply use and dispose shifted the social norm away from reuse and towards the prevailing throwaway society (König, 2019).
Reusable bottles are heavy and require additional effort to return, with consumers even preferring to pay the bottle deposit
rather than bringing the bottles back to the store (Eisenbach, 2004).

**R6: Market power of discounters – Lack of network effects – Undesirable social contagion.** Another pivotal development
was the strategic decision of discount stores to use mineral water to generate customer loyalty. With mineral water as an
established everyday product they were able to attract customers with an aggressive low-price strategy relying on single-use
bottles (UBA, 2010). Discounters had been rejecting the implementation of an infrastructure for reusable bottles to save staff
and logistical costs (storage space, return infrastructure, etc.). As a widespread reuse system only works well with
comprehensive industry collaboration, the increasing market dominance and influence of discounters, and their non-
participation in the system significantly undermined existing network effects and therefore the systems' effectiveness.
Furthermore, the immense listing of low-priced mineral water products resulted in brand manufacturers increasingly entering
low-priced distribution channels using single-use packaging (UBA, 2010; Stracke and Homann, 2017). This presumably led
to an undesirable social contagion effect, as the adoption of the more unsustainable alternative became more acceptable as
more companies followed suit.


**R7: Lack of performance – Lack of innovation – Lack of economic competitiveness:** The performance of the bottle reuse
system started to struggle as rising transportation costs, the repurchasing of new reusable bottles due to lower return rates, and
high storage space and staff costs for handling empties, had to be contrasted to lower infrastructure and material costs of single-
use bottles and more possibilities for differentiation of single-use bottles (UBA, 2010). Additionally, there was a persistent
decline in the financial and innovative capacity of reuse pools. Signs of a reinforcing downward spiral were indicated by
outdated reuse pool systems (both reusable bottles and crates) that caused an increasing number of companies to switch to
more contemporary, individual bottles. This in turn further destabilised the incumbent reuse pool system and increased its
obsolescence and shortcomings, that were not sufficiently addressed (UBA, 2010).

**Political change and policy intervention:** In the meantime, starting in the 1970s and with growing environmental movements
including the founding of the Green party, the issue of waste as an environmental problem gained significant attention in
Germany. Consequently, environmental politics increasingly became part of the agenda and new waste management laws were
introduced. After the German reunification in 1990, two different waste systems had to be aligned. While West Germany had
not introduced an infrastructure for the collection and recycling of glass and paper until the end of the 1970s, the efficiency of
the *SERO* waste collection system in East Germany was acknowledged as significantly higher than the collection system in
West Germany (UBA, 1992). Still, the intended preservation of the system after the German reunification could not be realised,
as the *SERO* system collapsed due to the huge volumes of waste from West Germany that were transferred to recycling sites
in East Germany, exacerbated by a credit fraud after the system was privatised (UBA, 1992). Instead, with the first German



Packaging Regulation in 1991 an Extended Producer Responsibility was introduced by the recently established Ministry of
Environment, making producers responsible for the waste management of the waste they produce (Quoden, 2010).

**Market intervention:** To avoid further regulation, the private sector in Germany organised a comprehensive second collection
system (today known as the *yellow bin or bag*) alongside the existing public waste disposal system, setting the conditions for
better organised recycling of packaging, but leaving any considerations for reuse aside. This so-called *Dual-System*, financed
by the 'Green dot' licensing (Seifert, 2011), also enabled the collection and recycling of single-use PET bottles, with relatively
advanced collection (80%) and recycling (66%) rates (IFEU, 2004). However, recycling rates for other types of packaging
remained rather low (Bünemann et al., 2011).

**Further policy intervention:** With the Packaging Regulation, the collection and recycling of single-use beverage packaging
was prescribed to remain integrated in the *Dual System* as long as the proportion of reusable beverage packaging did not fall
below the then current share of 72 %. Otherwise, according to the regulation, a mandatory deposit for single-use beverage
packaging would come into effect to make sure that the reuse market share did not decrease in light of the private collection
and recycling system in place (Hoffmann, 2011). However, already at the time of the second revision of the Packaging
Regulation in 1997/98, great concerns about the actual effectiveness of a mandatory deposit started to prevail through new
insights and policy learning, although it was still kept in the regulation for the time being (Hoffmann, 2011).

**R8: Increasing recycling efforts – Decreasing reuse efforts.** By implementing a waste collection system exclusively centred
around recycling and neglecting the integration of potential infrastructure for reuse, reuse efforts increasingly faded into the
background. While the *SERO* system demonstrated the possibility of reuse mechanisms for various materials, policymakers
directed their attention towards single-use recycling objectives, with the exception being their only apparent efforts to sustain
the last remaining packaging reuse system for bottles. The dominating presence of the comprehensive recycling system
reinforced the fallacy that its technology alone could effectively address the waste issue (Köster, 2021), causing alternative
waste reduction strategies to gradually lose relevance for politics, businesses and consumers.

**Business innovation and technology intervention: Introducing reusable PET bottles.** Eventually, the mineral water
industry could no longer ignore the advantages of the plastic bottle, especially since *Coca-Cola* marketed its beverages,
including its table water in 1988, in plastic bottles and consumers responded very positively to this packaging innovation
(Eisenbach, 2004). Overcoming challenges such as the development of taste-neutral PET granulates and resistance from
stakeholders, a leading mineral water company innovated a solution for individual reusable PET bottles in 1998 (Lippert et al.,
2012). The *GDB* cooperative followed up with the introduction of a reusable pool PET bottle and a matching crate in 1999
(Eisenbach, 2004). Based on current environmental assessment criteria (UBA, 2016) the reusable PET bottle, with its average




circulation rate of 25 times and lighter weight, surpasses the eco-efficiency of the reusable glass bottle, making it the most environmentally friendly beverage packaging option to date.

**R9: Urgency to act – Failed coordination – Lack of alternatives.** Since the aimed for reuse market share was still continuously missed, governmental authorities called for rapid measures. The government worked on an amendment of the Packaging Regulation in the beginning of the 2000s. They were no longer exclusively considering a mandatory deposit as the optimal tool as various alternative instruments had been deliberated and assessed. However, despite the widely recognised necessity for governmental intervention, it remained impossible to achieve consensus among federal and state government

bodies, industry experts, and academia regarding which alternative would prove most efficacious (Hoffmann, 2011).

According to Hoffmann (2011), the inability to reach consensus among stakeholders stemmed from the contrasting leadership styles of the Minister of Environment in 1998, Trittin, and his predecessor, Klaus Töpfer. Töpfer effectively coordinated negotiations by deprioritising his preferences and strategically supporting the Federal Council, leading to the successful

acceptance of the first Packaging Regulation and the acknowledgement of the impending mandatory deposit (Prüfer, 1999). In contrast, Trittin adopted a top-down approach, favouring a new amendment to the Packaging Regulation and warning of the implementation of the prescribed mandatory deposit if the amendment was not accepted. However, it was clear that he also opposed the old mandate, creating the perception that it would not be enforced even without an agreement on the amendment. This reduced the pressure to support the new amendment and explore alternative measures for reusing the system by both

businesses and other politicians (Hoffmann, 2011).

After the mandatory deposit was neither able to be efficiently designed in the phase of the policy formulation of the proponents, nor fully prevented by the opponents, it led to the implementation of a policy that was not desired by any party involved. The introduction of the mandatory deposit for single-use bottles in 2003 initially led to a new peak in reusable bottles, while single-

use bottles temporarily lost market share because retailers had not prepared an appropriate infrastructure, as they had not anticipated the mandate's actual implementation. But this was not to last.

**B2: Rebound effects of policy intervention.** After 2003 the reuse market share trend continued to decrease strongly, since the execution of the new deposit-return system mandate and subsequent amendments unintentionally reinforced the

inconvenience of reusable bottles, further destabilising that system, while incentivising single-use solutions: While deposited single-use bottles were made mandatory to be returnable at all retailers that sold beverages in the same material, this mandatory returnability has not been applied to reusable bottles. Many complementing technologies like vending machines made the DRS for single-use more efficient (König, 2019), but did not accept reusable bottles. Also, the higher deposit of single-use bottles (25 ct) compared to the deposit of reusable bottles (8–15 ct depending on bottle type) provided stronger incentives for the

consumer that reinforced the functioning of the single-use system. This undesirable, but by experts anticipated, rebound effect




of the mandatory single-use deposit (UBA, 2010) instead enabled and strengthened the stabilisation of the new single-use regime.

**R10: Lack of information – Lack of capability.** These regulatory developments, including the introduction of a mandatory
deposit system for single-use bottles in Germany, resulted in the existence of three parallel collection systems, reinforcing confusion and frustration among consumers and businesses: the household collection *Dual System* for recycling, the mandatory deposit-return system for single-use bottles, and voluntary deposit-return system for reusable bottles (Fig. 9). This complexity made it challenging to navigate the different collection processes and assess the differences between single-use and reuse, since it was incomprehensible which and why certain types of containers were subject to a deposit, with different deposit costs
depending on the size and type of packaging, while others were not deposited at all (Fig. 9). Additionally, consumer awareness of the differing environmental consequences has been limited, largely because the separate collection system for bottles creates the perception of adequate environmentally-friendly recovery, whether or not it is followed by recycling or reuse processes (GPPC, 2023).

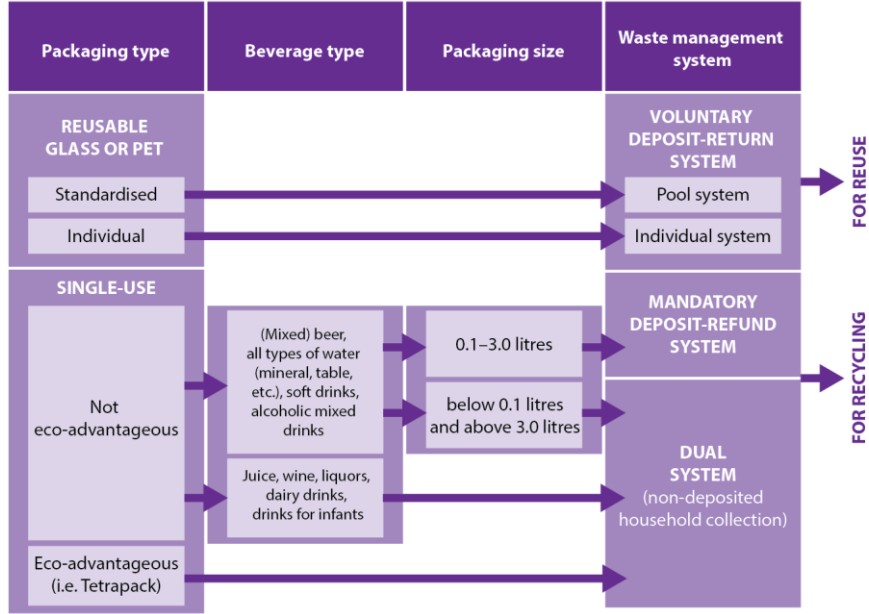

**Figure 9: Overview of current waste management systems for beverage packaging in Germany, adapted from PwC (2011).**

## 4.4 Current developments: 2010s–today

After the strong decrease in previous years, the market share for reusable bottles seems to have stabilised at around 40 % from 2010 till 2020, with indicators of a potentially increasing trend in the last few years. In the meantime, the *GDB* cooperative



has responded to current trends by introducing additional bottle sizes and designs. Currently more than 70 % of all reusable
bottles are *GDB* pool bottles (glass and PET) (GDB, 2023e), the rest are individual reusable bottles. Many established mineral
water companies and retailers offer water in several packaging types, aiming at different consumer segments, while most
discounters still exclusively offer single-use packaging. However, while LIDL relies on the alleged eco-efficiency of the bottle-
to-bottle recycling system (Kolf, 2023), ALDI recently announced it would restart testing a reusable bottle system from 2024
in light of the strongly increasing political interest in promoting circular strategies (Bender, 2023).

To improve the distinction between single-use and reusable bottles, the German government had planned on introducing
mandatory labelling on bottles in 2009. However, the EU commission rejected this, arguing it would impede the free movement
of goods in the internal market (Fachbereich Europa, 2016). This was followed by subsequent discussions and it was not until
the introduction of the new German Packaging Law in 2019 that the obligatory differentiation between "single-use" and "reuse"
labelling became mandatory on price tags at the point of sale (BMUV, 2019).

Currently central on the EU level is the Packaging and Packaging Waste Directive (PPWD) that "lays down measures to
prevent the production of packaging waste, and to promote reuse of packaging and recycling and other forms of recovering
packaging waste" (European Parliament, 2023). It sets out requirements that all packaging placed on the EU market must meet
with the objective to ensure that "all packaging is reusable or recyclable in an economically feasible way by 2030" (European
Parliament, 2023). The directive is currently revised and drafted into a regulation (PPWR). However, the reuse and refill targets
set up in the proposed regulation for beverage packaging is not as ambitious: Depending on the beverage type, the aimed for
reuse market share is between 5-10 % till 2030 and 15-25 % till 2040. Taking this as a benchmark, the market share in Germany
would already be more than sufficient.

In parallel, opponents of reuse systems lobby against the upcoming packaging regulation. The PPWR was described as the
"most lobbied on file" (Carlile, 2023). Industry members held over 290 official meetings with Members of European
Parliament on the topic between the beginning of 2022 and early April, compared to just 21 equivalent meetings held by NGOs,
while McDonald's and others have funded three studies, websites, and multiple articles attacking the legislation arguing it
would undermine Europe's net zero ambitions (Carlile, 2023).

Other ongoing discussions revolve around increasing the height of the deposit of reusable bottles, harmonising with the single-
use deposit (Boldt, 2023), the introduction of a reuse system for wine bottles, as well as making more beverage types subject
to the mandatory deposit (from 2024 dairy products will be included) (Effenberger, 2023).

This showcases the ongoing competing dynamics between the reuse and single-use recycling regime. While some argue that
making deposit systems for single-use recycling mandatory to deliver closed-loop circularity (NMWE, 2022), others are





convinced of the impeding lock-in effect of investing in more deposit-refund systems solely focusing on recycling that continue

525     the linear flow instead of incentivising reuse practices. Moreover, some pioneers are moving away from deposit schemes,

exploring fee-based systems particularly for reuse systems in the take-away food sector  instead (GPPC, 2023).

To sum up and to direct the focus on key interventions by business actors and policymakers in the following discussion, Fig.

10 provides an overview of the comprehensive historical timeline of our case.


**Figure 10: 1940–2020 timeline overview showing (from bottom to top) key business and policy interventions; circulation rates, circulations per year (pre-pool system introduction: expert estimates; post-pool system introduction: UBA (2022)) and market share of reusable bottles (UBA, 1983, 2022); German GDP  (Statistisches Bundesamt, 2023); German mineral water sales**



**(Eisenbach, 2004; Statista, 2023); revenue of the *GDB* cooperative (Eisenbach, 2004); members of the *GDB* cooperative (Eisenbach, 2004; GDB, 2023b). As indicated by the dashed lines, some data were not accessible to the authors.**

## 5 Discussion

Here, we discuss key learnings and opportunities for business and policy action to influence enabling conditions, reinforcing feedback loops and interventions that affect the speed of sustainable change, aligning with the PTP framework (Fig. 4) and contemporary insights.


### 5.1 Enabling and destabilising conditions

*(Lack of) Economic competitiveness: While the participation in the nationwide GDB pool system offered clear economic advantages in Phase 1, later in Phase 2 the established reuse system lacked the necessary resilience to remain competitive in the face of the advantages offered by emerging single-use packaging alternatives.*


If a system does not demonstrate a clear economic advantage over alternatives, a destabilisation of the system is likely. To tip the economics of sustainable systems a suitable policy mix, that recognises the external costs of the less sustainable alternative, is crucial and needs careful assessment. National and international standards are required for a defined pathway and clear framework for sustainable systems to provide a long-term vision and security for investors, addressing the current lack of 550 direction, that hinders venture capital investment (GPPC, 2023). Businesses, in turn, need to recognise and embrace the business opportunity of sustainable solutions and their long-term superiority (return on investment studies as from Peeters et al. (2023) can be particularly helpful for science-based guidance). A joint commitment by several businesses within an industry to invest in such solutions can reduce risks and costs and speed up the transition towards the new system (further elaborated in Sect. 5.2), especially if it is not yet backed up by policy support. Furthermore, comprehensive infrastructure investments 555 can contribute to the durability of sustainable change and the system's economic competitiveness, as they can create lock-ins to systems for twenty or more years (Tangri et al., 2022; Accorsi et al., 2014; Bauer et al., 2022; Klitkou et al., 2015).

*Enabling social network structure: The established social network structure by existing industry cooperatives simplified the collaboration to develop and reach a consensus on the pool system, subsequently boosting participation in the system.*


In our case study, creating an enabling social network structure by uniting various interest groups proved highly beneficial to find an effective market-based solution and to prevent undesirable policy interventions. Several national and international initiatives in the field of reuse, such as the *Mehrwegverband e.V.* in Germany, the *New European Reuse Alliance* or the *Global Plastics Treaty*, showcase promising current efforts for this approach. Additionally, the *Global Plastics Policy Centre* (2023) 565 emphasises the role of multinational companies as enablers. Given their extensive and complex supply networks, these





companies have the potential to play a pivotal role in coordinating and executing the infrastructure and logistics required for regional and global reuse systems within international supply chains.

Suitable governmental vision and leadership can help to incentivise and enable such a network structure. For example, early
recognition and addressing of the destabilising threat of non-participation of essential, inhibiting players (like discounters in our case) could be effective. If the market fails to collaborate, policymakers need to regulate to address the market failure. However, before rushing into policy interventions and creating lock-in effects of unproductive regulations, policy learning and experimentation guided by recent scientific insights on transformative innovation policy (Haddad et al., 2022) can help enforce businesses to action while leaving room for innovation.


*(Lack of) Performance, accessibility, and social norms: While the performance of reusable glass was initially superior to alternative single-use materials in terms of aesthetics, taste and feel for bottling water, the technological properties of single-use materials quickly improved and became competitive. Similarly, the pool bottle system first provided greater accessibility and convenience for the participating companies, but also simplified the return for consumers. This was later surpassed by*
*the convenience of the disposability of single-use packaging.*

Friction points for consumers must be minimised to optimise consumer convenience and behavioural adoption, as the normalisation of sustainable systems requires a substantial shift in consumer behaviour. In the case of reuse systems, there is a clear opportunity for businesses to find appropriate end-of-use system solutions that provide easy take-back options. These
do not necessarily have to be reinvented, but can take inspiration from historically proven solutions (e.g. the Milkman bottle take-back and reuse system (Vaughan et al., 2007) or the *SERO* waste management system (UBA, 1992). Further advancement of reusable packaging in other business models (catering, take-away, etc.) can complement and reinforce the establishment of reuse and return habits through learning-by-using processes (Arthur, 1989; Rogers, 1962). Additionally, policymakers can support the performance and accessibility of a system, for example, by enforcing mandatory take-back policies and by
preventing any confusion and subsequent dismissive consumer attitudes towards the return options through adequate educational measures.

*Lack of capability: The increased variety in bottles made it harder for consumers to differentiate between single-use and reusable bottles, causing confusion and frustration.*


To increase the desirability and legitimacy of a sustainable solution it is necessary to enable the capability to clearly understand the environmental benefits compared to the competing solution. In the case of reuse, the misconception that recycling alone is an adequate sustainable alternative is a major obstacle. By emphasising the hierarchy of waste reduction strategies (European



Commission, 2008), educational business and governmental campaigns can raise awareness and promote reuse systems,
contributing to a more positive public debate.

## 5.2. Key feedback loops affecting the speed of change

***Economies of scale and economies of reuse driven by standardisation:*** *Standardisation played an essential role in successfully transitioning to the widespread reuse system.*


Standardisation is recognised as a key driver for mainstreaming reuse systems in existing literature (Anastasiades et al., 2021; GPPC, 2023; WEF, 2021). It drives economies of scale and reuse, that in the case of bottle systems can lead to a return on investment after around ten years according to current findings (Peeters et al., 2023). Furthermore, the commitment to standardisation creates a lock-in that mitigates risks for businesses and establishes a well-defined pathway for the
implementation of scalable reuse systems, fostering innovation and offering assurance to.

As the standardisation of bottles is very straightforward with a limited range of formats, only requiring minor infrastructure changes, the transformation of already existing single-use return points to include reusable bottles should enable a fast and feasible transition. A likewise easy standardisation is applicable to packaging types ranging from jars for spreads, yoghurt and
soups to bottles for showering gel or cleaning products. Branding differentiation can still be achieved through labelling and colour choices, creating marketing opportunities within the established guidelines that align with the pooling infrastructure.

Generally, the early development of standards is substantial for an efficient and long-term approach to the introduction of reuse systems, not only for the packaging shape itself, but also the tagging, software and labelling to prevent isolated approaches.
The latter are more likely to have limited growth potential and higher rates of inefficiency beyond the localised boundaries of the reuse system (GPPC, 2023).

***(Missing) Network effects and coordination:*** *Coordination for a joint market-based solution can operate as an effective accelerator to shift a system in a direction that is widely desirable and accepted by all stakeholders, unlocking quick scaling*
*and adoption.*

If a large-scale entity (in this case the whole mineral water industry) together decides to commit to a specific system at the same time, the fundamental coordination game is simplified and the point where tipping becomes self-sustaining can be reached more quickly. Furthermore, our case exemplifies the importance of an entity or mechanism like the *GDB* cooperative
to initiate, facilitate and lead successful cooperation. A proactive initiator to spearhead collaborative efforts can bring industry stakeholders together and foster joint innovation. It allows for the pooling of expertise and resources, which is essential for



developing effective sustainable solutions. Finally, a collective commitment to substantial change is needed to ensure the prevention of possible competitive inequalities, which poses a common concern in the respective decision-making processes for sustainable transitions.


In contrast, the dominance of single-use packaging systems in phase two of our case exemplifies what happens if coordination efforts fail: If the involved interest groups do not prioritise and cannot agree upon investing in a common solution, even though it is urgently needed, positive change will be impeded. Moreover, this can ultimately lead to the risk of politics getting involved and, at worst, a flawed solution being implemented.


While the national bottle reuse pool system had clear economic advantages over the regional solutions beforehand, today's availability of competitive single-use packaging alternatives complicates the coordination game. Currently, ecological benefits often need to be weighed against economic disadvantages, although it is certainly possible that in the long-term reuse systems will dominate in terms of eco-efficiency in the face of emerging packaging regulations, unsteady material availability, rising

commodity prices, and waste management costs, contributing to reinforcing socio-ecological positive feedback. The adoption of reusable packaging is therefore no longer just an environmental imperative, but will increasingly be a necessity to maintain a profitable business (Kennedy, 2023; Peeters et al., 2023).

## 5.3 Business and policy interventions

While the successful introduction of the bottle reuse pool system exemplifies a well-executed business innovation and technology intervention (as described in Sect. 4.3) that stimulated the positive reinforcing feedbacks described above, interventions to stabilise the reuse market were insufficient to stop the rise of single-use bottles, and some even reinforced their rise.

The private investment into the introduction of a PET reuse pool bottle demonstrates the attempt of the mineral water industry to adapt to the market developments towards single-use plastic. Despite the increased convenience and innovative capabilities, the development of the reusable PET bottle alone has not been enough to stabilise the incumbent reuse system. It also increased the variety of bottle types, reinforcing the incapability to distinguish between reuse and single-use bottles. Thus, it failed as a balancing feedback mechanism to sustain reusable bottles as the prevailing packaging choice for beverages.


The policy intervention of introducing a mandatory single-use bottle deposit also failed by encouraging bottle recycling rather than reuse. This highlights the importance of enabling intelligent sustainable system design without shifting the environmental





burden, e.g. establishing a system that tackles the root of the problem in the long term, instead of creating competing systems that impede each other (Fig. 11).


To achieve this, and to maximise the benefits of circularity in system design, it is necessary to be mindful about different possible circular configurations. While recycling strategies for single-use packaging can reduce the waste problem (B1 in Fig. 11), it is essential to adopt a holistic perspective and re-evaluate their effectiveness. Single-use packaging can only be "recycled" (usually downcycled) for a limited number of cycles, wasting valuable resources in the process. As illustrated in our case, the wrong business and policy focus on one strategy can impede (R1) the realisation of unlocking the greater potential inherent in another (B2). Relevant insights and frameworks can help assess and explore circular configurations (Blomsma and Brennan, 2017; Blomsma et al., 2023).



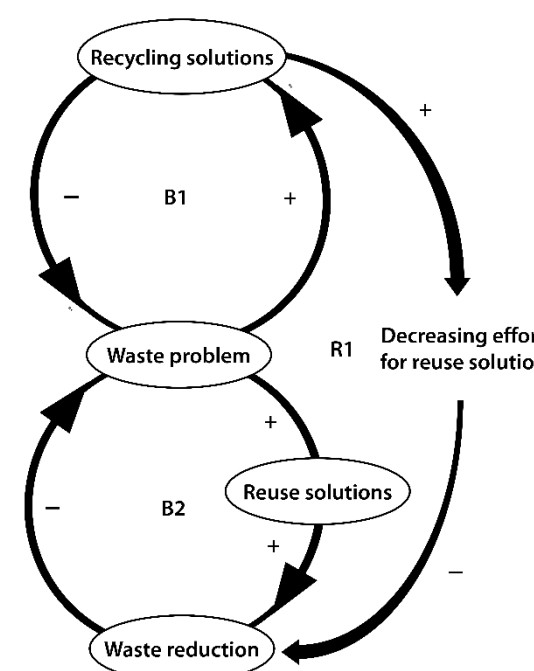

**Figure 11: Shifting the burden system archetype, recycling vs. reuse, adapted from Braun (2002)**

Today, some national policies as well as the current EU Packaging Regulation draft focus on reducing single-use plastic packaging by specifying reduction targets, that can be achieved through *either* reuse, recycling, or composting. However, allowing single-use options to meet these targets often results in efforts directed towards them rather than investment in reuse systems. Similarly, while bans on specific materials such as plastics can effectively address the problem of widespread littering in certain regions, they may inadvertently lead to substitutions that simply shift the environmental burden to the use of alternative single-use materials (Elisabetta Cornago et al., 2021), as linear approaches can never be truly sustainable (Break Free From Plastics, 2021). Thus, policy approaches with a heavy focus on recycling need to shift towards preventing the production of single-use packaging, focusing on improving and scaling up existing more sustainable solutions such as the bottle reuse system. Currently discussed policy measures are virgin material production caps and taxation, landfill and incineration taxation and binding reuse system targets (GPPC, 2023).



## 6 Conclusion


We have examined the positive and negative developments in a historical case study of the German bottle reuse system to understand tipping dynamics in sustainability transitions and the associated agency of businesses and policymakers. Numerous factors contributed to the successful shift towards the dominating reuse system, as well as to the failure to prevent tipping towards the competing single-use regime. We identified several enabling and destabilising feedback dynamics that were



influenced by landscape changes through historical and technological developments, and industry-specific business and policy interventions that accelerated or impeded change.

Key enabling dynamics to the successful transition to the widespread bottle reuse system were the superiority of the well-designed standardised pool bottle solution and industry-wide collaboration. However, the subsequent tipping towards a single-use packaging regime was not prevented by business or policy interventions. While several drivers of this transition were shaped by market changes and external global and technological developments that may have been beyond direct control or influence, certain aspects, such as the rebound effects of introducing a mandatory deposit-refund system for single-use bottle recycling, the failed coordination to find an alternative solution, and over-complexity of the resulting diversity of bottle waste management systems, contributed to the negative tipping.

Based on our findings and in line with the PTP framework (Fig. 4), we specifically highlight the importance of economic competitiveness, social network structure, performance, accessibility, social norms, and capability for enabling a sustainable transition to reuse systems. Further, we identified economies of scale and reuse driven by standardisation, and network effects as essential reinforcing feedbacks, that determine the speed of change.

Finally, when it comes to effective interventions, we recognise the importance of considering intelligent system design without shifting the environmental burden, to make sustainable change last. Both linear and circular paradigms will likely coexist for several more years, suggesting that the effective management of competing dynamics between different strategies and systems is essential to prevent undesirable lock-in effects.

There are several opportunities for future research, recognising that suitable strategies to establish and scale reuse systems in other packaging contexts will depend on the sector. Venues, event and on-site dining versus on-the-go eating and drinking, e-commerce, fast-moving consumer goods or business-to-business are all potential areas for further investigation. Additional historical case studies on the establishment of waste management systems could be explored to guide decision-making in upcoming national and international packaging regulations. Finally, other key actor perspectives should be investigated, that we only superficially touched upon, such as the importance of consumer behaviour, transportation and logistics in the supply chain, or multinational companies for global scalability.



*Author contributions.* MO designed the study, conducted the research and wrote the paper with input from FB and TL. FB and
MO prepared the figures with input from TL. TL contributed the theoretical framework and edited the paper.

*Competing interests.* At least one of the authors is a member of the editorial board of Earth System Dynamics.

*Acknowledgements.* The authors would like to thank Tobias Bielenstein for an insightful interview and Dr. Uwe Spiekermann
and Dr. Stefanie van de Kerkof for helpful advice on economic history research.

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
