# Peer review of ""History in a bottle:" Tipping dynamics in packaging systems - the"

_EGUsphere, 2023_

## Author Response (AR1)

**Author's response**

**Referee's comment #1**

The paper presents an interesting case study for Lenton et al.'s positive social tipping point framework.

The manuscript is well structured in the beginning but in total, too lengthy. Especially part 4 and 5 could be shortened and lack clarity in several aspects. It is for example not entirely clear why reuse systems are better than recycling systems. What are the measures of a 'good' system? Economic efficiency, the carbon footprint or the return/recycling rate? A clearer definition would help the reader to understand the framework.

In more detail:

20

The problem of a decreasing share of reusable glass bottles it not clear to me when the alternative is a plastic bottle recycling system. According to this study, e.g., recycled PET is more sustainable across a range of indicators compared to reusable glass: Stefanini et al., 2021. Plastic or glass: a new environmental assessment with a marine litter indicator for the comparison of pasteurized milk bottles.

Given that the advantage of reuse glass bottles over recycling PET is not clear, arguments in the discussion section might also not hold true. 'Policymakers need to regulate to address the market failure' regarding reuse system could become a misguided policy intervention if the assumption that reuse is better than recycling is not certain anymore.

Figure 3: Does ESD accept previously published figures?

Figure 5: What's shown on the x-axis? And how is stability defined?

Figure 6 is confusing in several aspects:

- The axes are missing, especially the y-axis. It is also not clear what reinforcing and balancing refers to here.
- In the example, R2 (innovation and technology intervention) happens before the social contagion effects, which is contrary to what you describe in Fig 4.
- How can a reinforcing and balancing feedback loop have the same direction on the y-axis?

Figure 8: similar to Fig. 6, the axes are missing. While it seems that the x-axis (time) is consistent between the two figures, the y axis is changing which is confusing to the reader. In addition, in Fig 6 the downward trend seems to be something desirable in terms of efficiency and sustainability (or whatever the authors intend to show here), whereas in Fig. 8, the downward trend seems to be something 'negative'.

R6: How can network effects (R3 in Fig 6) and a lack of network effects both be a reinforcing feedback loop?

R7: Similarly, it is counterintuitive that the lack of something is a reinforcing feedback mechanism.

I recommend using one single variable for the y-axis in Fig 6 and 8, which corresponds to the x-axis in Fig 5.

As statistics of reuse glass bottle and recycling PET bottle use seem to be available, the figures would gain significant power if the shares were shown quantitatively in the figures to demonstrate tipping effects over time.

It does not make sense to describe B2 before R10 if R10 happens before B2 according to fig 8.

In the discussion part, the difference between 5.1 (enabling/destabilizing conditions) and 5.2 (feedback loops) is not clear. In general, the discussion section could be shortened, given the great length of the manuscript. Policy options in 5.3 are partly covered in 5.1 already and it is not clear why an additional section on business and policy options is needed.

Overall, the empirical example of glass bottle reuse systems in Germany is an interesting illustration of Lenton et al.'s PTP framework but it requires considerable improvements, especially regarding the clarity of the figures.

**Authors' response:**

45

Thanks for your interest in this manuscript and your detailed review, which helped to improve it significantly. Below, we respond to your comments and suggestions one by one, where the paragraphs in cursive are your original comments:

The paper presents an interesting case study for Lenton et al.'s positive social tipping point framework. The manuscript is well structured in the beginning but in total, too lengthy. Especially part 4 and 5 could be shortened and lack clarity in several aspects.

RE: Thank you for the invested time and effort in providing the valuable and much appreciated feedback. We can see the point the reviewer is making. As with historical cases, this case is very rich and there is much to say about it. However, we will shorten our discussion of the case narrative (section 4 - results) by means of providing a short introduction and condensing the remainder into a table accompanying the images of both tipping phases. This way, we can move some of our observations about the insights into tippings this generates from section 5 (discussion) to section 4. At the same time, in section 5, we can focus more on the relationship of tipping points with the traditional timelines reserved for such transitions and choosing time appropriate timescales to study change (i.e. although tipping itself in this case happened relatively fast, it seems that other slower changes also played a role) and the role of interventions that initiate tipping (i.e. how specific interventions managed to bring about a sudden acceleration). Particularly the latter raises an interesting question for further research: if and how 'tipping' actions can be recognised or designed given a certain set of circumstances. We are exploring if this can be linked to other concepts such as 'the adjacent possible theory' by Kaufmann to provide concrete directions for further work (e.g. how to recognise the constraints of the situation and use this as a basis for interventions). All in all, we think that these changes will both shorten sections 4 and 5 as well as deepen the contributions from this study.

It is for example not entirely clear why reuse systems are better than recycling systems. What are the measures of a 'good' system? Economic efficiency, the carbon footprint or the return/recycling rate? A clearer definition would help the reader to understand the framework.

RE: The reviewer has pointed out something that we see can cause confusion. Please let us clarify: our study focuses on understanding the dynamics of tipping from one paradigm (or one system state) to another. The quantitative assessment of which of these states is the "most sustainable" we consider out of scope for this work, as it is highly context dependent, the domain of active scientific inquiry, and our paper does not focus on Life Cycle Assessment or the quantification of environmental impact. Although, as we mention in the paper, there is evidence that points to the system under study comparing favourably to single use systems (based on circulation rate and distance travelled), and it is also the assumption of many current policy interventions in the circular economy domain that aim to increase reuse rates. Our objective instead is to focus on understanding tipping dynamics in a qualitative manner and we intend to revise the manuscript to highlight this better.

In more detail: The problem of a decreasing share of reusable glass bottles is not clear to me when the alternative is a plastic bottle recycling system. According to this study, e.g., recycled PET is more sustainable across a range of indicators compared to reusable glass: Stefanini et al., 2021. Plastic or glass: a new environmental assessment with a marine litter indicator for the comparison of pasteurized milk bottles. Given that the advantage of reuse glass bottles over recycling PET is not clear, arguments in the discussion section might also not hold true. 'Policymakers need to regulate to address the market failure' regarding reuse system could become a misguided policy intervention if the assumption that reuse is better than recycling is not certain anymore.

80

RE: The rapid establishment and almost equally fast undoing of the initially successful change described in our case study is relevant for current circular economy change efforts that are the focus of many current environmental policies, the EU among them. Many rapid transitions are pushed for in this area, including packaging. Here, the objective is often to move away from recycling and towards increased reuse. However, the reception is not always positive and these efforts so far do not always show the desired results: there is critique on the usability of solutions, on how this – perversely – leads to increased plastic usage, and how return rates fall behind expectations. This undermines reuse systems and it raises the question of whether they can be expected to be permanent or that they will – like in our case – be replaced by single-use systems once more. Given the current investment of effort and resources in reuse systems it therefore seems prudent to learn from similar efforts in the past to see if lessons can be learned that can inform these current change efforts; hence the choice of our case study. However, as

90 mentioned in our reply to the previous comment, we regard the environmental impact assessment to fall outside of the scope of our study. In this work we are interested in the tipping dynamics themselves, and by unpacking and better understanding this to contribute both to a better understanding of tipping dynamics as well as circular economic change efforts. As previously stated in the above, we intend to revise the document to make this focus explicit. In addition, it seems prudent to, also in the discussion and conclusion section, once more point out that the quantification of environmental impact is not included in this work - and that other or further work is needed to establish under which conditions reuse systems are also environmentally favourable. We will add that it is here, at the intersection of our work and these quantification efforts, that a more comprehensive picture can be built about reuse systems, how to bring them about and under what conditions they are environmentally favourable.

Figure 3: Does ESD accept previously published figures?

100

105

110

115

RE: Yes, as long the reproduction rights are secured, which is the case here. However, we intend to align the images so they are presented in the same style in the final document, which entails redrawing the existing figure.

Figure 5: What's shown on the x-axis? And how is stability defined?

RE: We will change the figures in question so that the x-axis is labelled as the "level of reuse". From lowest to highest level of reuse: Single-use recycling, individual company reuse, pool reuse. (Dynamic) stability in this context refers to the ability to resist perturbation and recover from it, which is visualised as steepness of the valley sides and the height of the hill that has to be overcome. We will include this more clearly in the text, already where we explain the positive tipping points framework, so it is clear to the reader when subsequent images are introduced.

Figure 6 is confusing in several aspects:

- The axes are missing, especially the y-axis. It is also not clear what reinforcing and balancing refers to here.
- In the example, R2 (innovation and technology intervention) happens before the social contagion effects, which is contrary to what you describe in Fig 4.
  - How can a reinforcing and balancing feedback loop have the same direction on the y-axis?

RE: Many thanks for your careful scrutiny of the images and your attention to detail. We agree that this figure needs to be revised. Firstly, the green arrows in this version are inadvertently mislabelled as 'destabilising' dynamics - they should be 'enabling' dynamics. Apologies for any confusion this caused.

The reinforcing feedback loops refer to the dynamics that reinforce the tipping towards a new system. The balancing feedback loops refer to the dynamics that stabilise the new system. We will make this clearer in the text in section 3 where we will say more about how we analysed the data and the key of the figure.

125

130

135

R2 refers to "economies of scale and reuse", which is in the correct order in the figure. The innovation and technology intervention you refer to is not shown in the figure, as interventions have been excluded from the figure to reduce complexity and visual clutter. However, we see the value in adding those details and will include all identified enabling/destabilising conditions/dynamics, feedback loops and interventions and will add these concrete interventions to the image so that they have a clear place in the timeline, yet do not confuse the tipping dynamics overview.

Figure 8: similar to Fig. 6, the axes are missing. While it seems that the x-axis (time) is consistent between the two figures, the y axis is changing which is confusing to the reader. In addition, in Fig 6 the downward trend seems to be something desirable in terms of efficiency and sustainability (or whatever the authors intend to show here), whereas in Fig. 8, the downward trend seems to be something 'negative'.

RE: Y-axis is still "stability" as in Fig. 5, both axes will be labelled accordingly.

'Down' in the tipping points framework merely represents a new system state. For tipping phase 1, we have chosen to represent the change from the individual company reuse system to the pool system as a change from left to right and from up to down to indicate the dynamic nature of the 'valleys' and 'walls' of system states. For tipping phase 2, to emphasise a reversal or undoing of the pool system we have chosen to represent it as a reverse movement compared to the tipping phase 2. We will emphasise this better in both the textual explanation of the tipping diagrams as well as in the visuals.

145 R6: How can network effects (R3 in Fig 6) and a lack of network effects both be a reinforcing feedback loop? R7: Similarly, it is counterintuitive that the lack of something is a reinforcing feedback mechanism.

RE: A reinforcing feedback loop can operate to amplify change in either direction. Network effects and Social contagion reinforced the tipping towards pool reuse in phase 1 (the more companies cooperated, the more attractive it became for others

to join), while the lack of network effects reinforced the tipping towards single-use recycling in phase 2 (the fewer companies participated in pool reuse, the less attractive it was to join the pool system). Reinforcing (positive) feedback can always operate to amplify change in either direction. In one direction increasing network effects leads to further increasing network effects.

In the other direction breakdown of network effects causes further breakdown of network effects.

155 I recommend using one single variable for the y-axis in Fig 6 and 8, which corresponds to the x-axis in Fig 5.

RE: We will keep the y-axis consistent for "stability" and the x-axis for "levels of reuse" in the figures.

As statistics of reuse glass bottle and recycling PET bottle use seem to be available, the figures would gain significant power if the shares were shown quantitatively in the figures to demonstrate tipping effects over time.

RE: Thank you for this input, we will do so where available.

It does not make sense to describe B2 before R10 if R10 happens before B2 according to fig 8.

RE: Right, we will rework this so that the order of the dynamics in the text fits accordingly to the figures.

In the discussion part, the difference between 5.1 (enabling/destabilising conditions) and 5.2 (feedback loops) is not clear. In general, the discussion section could be shortened, given the great length of the manuscript. Policy options in 5.3 are partly covered in 5.1 already and it is not clear why an additional section on business and policy options is needed.

RE: We agree, and will revise as indicated in the above. This means that we will rework and restructure the discussion section to be less repetitive, clearer and shorter.

**Referee's comment #2**

160

165

170

Overall, based on a sustainable transition perspective, the paper provises an interesting case study on historical account of rise and fall of a bottle reuse system in Germany. The paper in general is well-written and clear in its description. At the same time the reviewer felt that the chapters 4 to 6 are too lengthy. Especially, the discussion in the chapter 4 tends to be a descriptive account of historical development of the bottle reuse system rather than an analysis on how sustainable transition was

established and failed (or taken over by another system). Framework presented in figure 4 is reasonable. However, the storyline of connecting chapter 4 and 5 via the framework presented in Figure 4 is rather weak.

**Authors' response**

180

185

190

Thanks a lot for the interest in our manuscript and your feedback. As indicated in the other comment reply above, with historical cases, this case is very rich and there is much to say about it. However, we will shorten our discussion of the case narrative (section 4 - results) by means of providing a short introduction and condensing the remainder into a table accompanying the images of both tipping phases. This way, we can move some of our observations about the insights into tippings this generates from section 5 (discussion) to section 4. At the same time, in section 5, we can focus more on the relationship of tipping points with the traditional timelines reserved for such transitions and choosing time appropriate timescales to study change (i.e. although tipping itself in this case happened relatively fast, it seems that other slower changes also played a role) and the role of interventions that initiate tipping (i.e. how specific interventions managed to bring about a sudden acceleration). Particularly the latter raises an interesting question for further research: if and how 'tipping' actions can be recognised or designed given a certain set of circumstances. We are exploring if this can be linked to other concepts such as 'the adjacent possible theory' by Kaufmann to provide concrete directions for further work (e.g. how to recognise the constraints of the situation and use this as a basis for interventions). All in all, we think that these changes will both shorten sections 4 and 5 as well as deepen the contributions from this study.

---

## Author Response (AR2)

Dear Prof Otto and kind reviewer,

Many thanks for this opportunity to further clarify and improve our work. With this submission, we feel we have made another step forward.

Please see our responses to the comments and queries below.

We look forward to your feedback.

With warm regards, The Authors

**REVIEWER COMMENTS & OUR RESPONSES**

The revised version is a completely re-written manuscript rather than a reply to the reviewer's comments. Track change mode was only used in parts of the manuscript which made it difficult to recognize changes.

We apologise for this error in incompletely tracking the changes. This time around we clearly indicate all changes in content and terminology: with new and added text in blue and a strikethrough for deleted text. (Some very small edits related to spelling and grammar we don't always indicate to keep the revision document as simple as possible). We believe that addressing some of the comments in the previous review round warranted restructuring the paper - and that this has resulted in a more compact and clearer manuscript. Many apologies if we did not convey this sufficiently clearly in the previous submission in our letter and response.

**We group the following comments:**

- Comments were only partially addressed and important questions such as the suitability of reuse bottle systems as 'positive tipping element' remain open.
- The tipping framework used in this manuscript differs from the cited tipping point framework by Lenton et al as (i) Lenton's framework always refers to 'positive' in the sense of more sustainable while the manuscript partly describes a negative social tipping point,

We agree that the second tipping episode is a step backwards with regards to having a positive impact and that a name change to 'negative social tipping point' is more appropriate. We have made this change in the document and in the figures.

We have grouped the following comments due to a common cause or concern expressed by the reviewer for clarity and brevity:

- (ii) terms such as balancing feedback loops are used differently across the two frameworks. This deviation needs to be made very clear in the text.
- RE: Figure 6: The authors state that 'The balancing feedback loops refer to the dynamics that stabilise the new system.' Lenton et al (2022) or the Global Tipping Point Report, use a different definition: balancing feedback loops stabilise the old system. From Lenton et al. (2022): 'negative feedbacks maintaining the initial state', 'Existing regimes, whether social, technological or ecological, are stabilised by damping feedbacks that resist change and restore the status quo' or 'balancing negative feedbacks maintaining the initial state'. A 'balancing feedback loop' as a dynamic that stabilizes a new steady state can be introduced, but then it needs to explain that this is a new addition to the existing PTP framework and that it differs to the existing understanding of 'balancing feedback loops'.
- From Lenton et al. (2022): 'To bring a system to a tipping point typically requires some forcing that is, a change in boundary conditions in a direction that weakens balancing negative feedbacks maintaining the initial state and/or strengthens reinforcing positive feedbacks that amplify change (Meadows, 1999, 2008).

Balancing (or mathematically 'negative') feedback loops simply act to maintain the current state (attractor) of a system regardless of whether it is a desirable or undesirable state of the system or whether it is the old state or the new state - i.e. balancing feedback loops stabilise both the old state and the new state of a system, but they can be different balancing feedback loops in each case. Previous work emphasised the role of balancing/negative feedback loops in stabilising the initial state (and of amplifying/positive feedback in propelling change) but did not elaborate the role of balancing/negative

feedback in stabilising the new state. Here we elaborate on that (and how the balance of feedback shifts through different phases) because it is essential to understanding the overall chronology and multiple episodes of tipping in our case study. This is not a change in the well established terminology and widely accepted meaning of damping/balancing/negative feedback in cybernetics and systems thinking. But it is an elaboration of the overall positive tipping points framework that we now make explicit. It necessitates clearly identifying and differentiating balancing feedbacks that stabilise the old state and the new state which we now do more clearly in our figures and narrative.

-> Balancing feedbacks are exclusively seen as something undesirable that prevents 'positive'/desirable change.

This is an incorrect statement and a misinterpretation of the previous work on PTPs (which may have been insufficiently clear). It is important to stick to the widely accepted and more general (mathematical) meaning of balancing/damping/negative feedback loops, which refers to a general dynamic that keeps things in place/ stable. We have sought to clarify that in our revisions.

If the authors use 'balancing' as something so fix the new steady- state (e.g. something desirable as pool reuse systems), they need to make it very clear in the text that they deviate from Lenton's and the PTP community's definition of 'balancing'. If the authors followed Lenton et al (2022), an example of balancing feedbacks loops would be on p. 17, l. 420: lack of leadership and difficulties to align prospective partners as balancing feedback loop that prevents a tipping towards a more sustainable reuse pool system.

We are not deviating from our previous work or widely accepted definitions in systems thinking, although we are elaborating the positive tipping points framework somewhat. Given the confusion the reviewer is expressing we have chosen to carefully elaborate on the role of balancing feedbacks on both sides of tipping (pre- and post-tipping). That is: already in section 2.3 we indicate clearly that we regard destabilisation to be caused by the weakening of feedback loops that keep the current system in place, but that balancing loops also have a role post-tipping to keep the new systems state in place, and we have brought the remainder of the document in-line with this.

From the abstract: "Building on current research on positive tipping points, our case study demonstrates opportunities to create an environment for change, the role of reinforcing feedback loops in accelerating sustainable transitions, and successful interventions. However, the case also demonstrates the threat of destabilisation of newly created systems as a result of the emergence of competing technologies, in this case single-use plastic bottles. Unsuccessful efforts to stop this, included the introduction of a reusable plastic bottle and a failed policy intervention that rushed into a solution that instead accelerated the change it was designed to prevent."

-> The authors frame this as two separate tipping processes. An alternative interpretation of the German bottling system could be that recyclable single-use plastic bottles are a "shallow outcome", following the update of Lenton et al' (2022) framework in the Global Tipping Point Report (2023), section 4, p. 14.

We find it insightful to highlight two separate tipping episodes. Especially given that Episode 2 is not an improvement when it comes to environmental impact, but a step back. With the remainder of the changes (see also below), we trust we have sufficiently addressed this.

**We have grouped the following comments:**

- Regarding re-use versus recycling systems: The authors refer to the 'positive' tipping points framework by Lenton et al. Positive tipping in this framework refers to a transformation towards a sustainable future. 'Positive tipping points offer hope for accelerating change to avert climate and ecological emergency', state Lenton et al (2022). The Global Tipping Point Report (2023) defines on page 8, part 4: 'What is considered normatively 'positive' or beneficial, and by whom, is highly debatable. In principle, tipping points may be considered positive either: a) where they reduce the drivers of 'negative' Earth system impacts such as greenhouse gas emissions or deforestation, for example in a rapid shift to renewable energy or alternative food proteins; or b) where they improve the social foundations of sustainability.
- As such, the authors' statement that 'the quantitative assessment of which of these states is the 'most sustainable' we consider out of scope for this work' is worrying given that they use the 'Positive Tipping Framework' by Lenton as basis of the study. This question needs to be addressed in a short discussion of what the study defines as 'positive' and why the re-use bottle system is a valid example for a positive tipping element.

- In the authors' answer to my comment on Fig 5, I take that the 'desirable' state of 'better' is the level of reuse. That makes sense, but again it needs a short discussion why pool reuse of glass is 'better'/more desirable than recyclable PET.
- RE: The rapid establishment... > Again, the authors have not addressed my comment which is a link to the Stefanini paper and the question whether reuse systems are a useful example for positive tipping processes.

Taking the comments above together: we acknowledge that our response in the letter was somewhat different from what was in the final manuscript text. That is: in the manuscript we do not refer to "quantitative assessment" explicitly. However, the essence of our argument, as explained in the third paragraph of section 3.1, is that the number of reuse cycles and the distance traveled (40-50 uses and 260 km), far exceeds the break-even point (3-10 uses and 500 km) where reuse is thought to be more sustainable than recycling, according to Coelho et al., 2020; DUH, 2014b, EMF, 2023 and UBA, 2016 - as referenced in the document. It can therefore be expected that this system has less environmental impact than single-use alternatives. We therefore believe it is a reasonable assumption to frame the first tipping episode as a Positive Tipping Episode, especially in relation to the later introduction of plastic single-use and recycling. We do, however, regard doing a (retrospective) LCA or similar assessment as outside the scope of this study. To avoid confusion, we have removed the footnote, and added a sentence to the paragraph to make explicit that the case falls well within the break-even limits in the third paragraph of section 3.1. We have also amended the titles of the images: where Fig.4 is now clearly labelled as a "positive tipping point: and Fig. 5 as a "negative social tipping point."

**RE: A reinforcing...**

Again, there is a fundamental difference between Lenton et al.'s (2022)/the Global Tipping Point Report's (2023) and the authors' interpretation of 'positive tipping points'. This needs to be made clear and well described at some point in the paper. In their reply, the authors state that 'a reinforcing feedback loop can operate to amplify change in either direction.' This is not the case in how Lenton et al define 'reinforcing'. There, 'reinforcing' always refers to a positive (e.g. more sustainable) change.

There is not a fundamental difference, although our previous work may have been insufficiently clear. While the examples of reinforcing feedback given in previous studies may have always been in the normatively 'positive' direction, reinforcing feedback is simply an alternative label for mathematically 'positive' feedback and mechanistically it can always operate in either direction. (For example, just as there are 'increasing returns' to increasing adoption, there are decreasing returns to abandonment. To be more specific: the more people who abandon petrol/diesel cars and the more petrol/diesel stations close and mechanics retrain, the more inconvenient it is to (still) own a petrol/diesel car - and thus the stronger the incentive to abandon the technology (a reinforcing feedback).) Thus reinforcing feedback cannot be conflated with normatively 'positive' change, it has to be retained as a more general mechanistic concept. Normative 'positive' and 'negative' judgements can be assigned to the different directions in which a particular reinforcing feedback is operating, and that is what we now try to do in a clear and consistent way in the manuscript.

Research papers should start with an introduction into a topic. The re-written version does not provide a proper introduction but jumps straight into the research question without providing any contexts.

We apologise for that oversight and have now added a new first paragraph to the paper that clearly introduces the topic of circular economy and its relevance.

**p.3, I. 80. How does today's situation regarding soap bars compare to the historic re-use bottle system case study?**

We have opted to drop this example as it was causing confusion - it was intended to indicate that the solution space is not singular: e.g reuse is not the only option for introducing more sustainable options. In some cases, the product itself can also be redesigned or reformulated, realising further environmental benefits. But both reusable bottles and soap bars were once replaced by other more resource intensive alternatives.

p.10, I.251 The paragraph is confusing and was clearer in its initial form. 'Thus far, the framework has been applied to the food and land use system' sounds as if FOLU was the only application of the PTP framework so far which is incorrect. And how have food and land use systems 'provided initial insights into the adoption of renewable energy and electric vehicles'? Apologies, this was a mistake. We have now altered the sentence and included reference to Meldrum et al. (2023) on the energy system and to the Global Tipping Points Report 2023.

Fig. 4. Again, it should be mentioned that R1 and R2 follow a different logic than Lenton et al. (2022). Following Lenton et al, R1 would be a weaking of balancing feedbacks (balancing as keeping the old bottle system and weakening would refer to the loss of these bottles in WW II). The same applies to R2. R5 belongs to the next graphic as it has nothing to do with the tipping from company bottles to pool reuse.

Thanks for these excellent suggestions. We agree that R1 and R2 were wrongly described. The figure has been updated accordingly. However, we have preserved R5 as these developments already take place early in the timeline: it indicates that already early on the possibility of tipping in a different direction existed, and it emphasises the overlap of the two tipping episodes when it comes to certain developments. Also, R5 and B4 interact as described in the narrative (post-tipping for Episode 1) - so it is appropriate to mention it here. And: it also shows that the risk of single-use already loomed early and that such signs can be taken as early warning signs.

Fig 5. The entire example would not be defined as 'positive tipping point' following Lenton et al (2022) or Otto (2020) but rather be a 'negative social tipping point' as (according to the authors, see comment on sustainability above) PET reuse systems are less sustainable than glass reuse bottles.

We agree (and never thought of this second tipping episode as 'positive') - it was a negative tipping point from reusable glass to single use recycled plastic bottles and now clearly labelled as such.

p. 32, I. 758. Again, there is a reference to PTPs that assume a shift to a more sustainable state following Lenton which, according to the authors, is not the case in Fig. 5

This was a mistake that has now been corrected.